# RNA Polymerase II “Pause” Prepares Promoters for Upcoming Transcription during *Drosophila* Development

**DOI:** 10.3390/ijms231810662

**Published:** 2022-09-13

**Authors:** Marina Yu. Mazina, Elena V. Kovalenko, Aleksandra A. Evdokimova, Maksim Erokhin, Darya Chetverina, Nadezhda E. Vorobyeva

**Affiliations:** 1Institute of Gene Biology, Russian Academy of Sciences, 119334 Moscow, Russia; 2Center for Precision Genome Editing and Genetic Technologies for Biomedicine, Institute of Gene Biology, Russian Academy of Sciences, 119334 Moscow, Russia

**Keywords:** transcription, RNA polymerase II, Pol II pausing, Pol II stalling, poised Pol II, *Drosophila*, development, transcription regulation, NELF, DSIF, PAF, Brd4/Fs(1)h

## Abstract

According to previous studies, during *Drosophila* embryogenesis, the recruitment of RNA polymerase II precedes active gene transcription. This work is aimed at exploring whether this mechanism is used during *Drosophila* metamorphosis. In addition, the composition of the RNA polymerase II “paused” complexes associated with promoters at different developmental stages are described in detail. For this purpose, we performed ChIP-Seq analysis using antibodies for various modifications of RNA polymerase II (total, Pol II CTD Ser5P, and Pol II CTD Ser2P) as well as for subunits of the NELF, DSIF, and PAF complexes and Brd4/Fs(1)h that control transcription elongation. We found that during metamorphosis, similar to mid-embryogenesis, the promoters were bound by RNA polymerase II in the “paused” state, preparing for activation at later stages of development. During mid-embryogenesis, RNA polymerase II in a “pause” state was phosphorylated at Ser5 and Ser2 of Pol II CTD and bound the NELF, DSIF, and PAF complexes, but not Brd4/Fs(1)h. During metamorphosis, the “paused” RNA polymerase II complex included Brd4/Fs(1)h in addition to NELF, DSIF, and PAF. The RNA polymerase II in this complex was phosphorylated at Ser5 of Pol II CTD, but not at Ser2. These results indicate that, during mid-embryogenesis, RNA polymerase II stalls in the “post-pause” state, being phosphorylated at Ser2 of Pol II CTD (after the stage of p-TEFb action). During metamorphosis, the “pause” mechanism is closer to classical promoter-proximal pausing and is characterized by a low level of Pol II CTD Ser2P.

## 1. Introduction

The development of a multicellular organism occurs due to the timely activation of genes in cells of a certain type, which develop in the form of “transcriptional waves” [1,2]. The transcriptional cycle of genes in multicellular organisms is under the control of both temporal and spatial regulators. Its progression requires the clear coordination of various regulatory systems.

The RNA polymerase II enzymatic complex contains 12 subunits that are highly conserved from yeast to humans [3]. Metazoan (and *Drosophila* in particular) RNA polymerase II contains an additional stoichiometric, tightly interacting Gdown1/POLR2M subunit [4]. The progression of RNA polymerase II through the various stages of the transcriptional cycle is closely associated with a modification of the CTD domain of the Rpb1 subunit (Pol II CTD) [5]. Evolutionarily conserved Pol II CTD is composed of a species-specific number of repeats of the consensus amino acid heptad YSPTSPS (which, in *Drosophila*, are represented by 45 repeats, of which only two correspond exactly to the evolutionarily conservative consensus) [6]. An unphosphorylated state of Pol II CTD is important for interaction with the Mediator and the formation of the preinitiation complex (PIC) at the promoter [7]. Phosphorylation of Pol II CTD Ser5 by TFIIH is linked to promoter escape and mRNA capping [8,9]. Phosphorylation of Pol II CTD Ser2 by p-TEFb (consisting of cdk9 and cycT) is associated with the progression of the enzyme into the gene body, productive elongation, and the recruitment of splicing and polyadenylation factors [10]. At all stages of transcription, Pol II CTD acts as a platform that ensures the interaction of Pol II with a variety of complexes necessary for transcription [11]. Recently, RNA polymerase II CTD phosphorylation was found to play an important role in the transition of the elongating enzyme between the phase-separated condensates responsible for transcription initiation and RNA processing [12]. Despite the fact that *Drosophila* has highly divergent CTD repeats, they are also actively phosphorylated at Ser2 and Ser5, which makes all of the above described information relevant [6].

The multistage nature of the transcription cycle allows its various stages to be used for regulatory purposes. A large portion of inactive Metazoan genes are in a state of transcriptional “pause” when RNA polymerase II is recruited to gene promoters but no productive transcription is observed [13]. It has been suggested that this state allows genes to be controlled simultaneously by various regulatory systems [14]. Indeed, when different steps of the transcription cycle (e.g., initiation and elongation) are stimulated by different regulators, efficient transcription is possible when all the types of regulators are active. This hypothesis was confirmed using the experimental model of heat shock-activated genes, which are controlled by both GAF and HSF1 transcription factors [15]. It was shown that the former stimulates the recruitment of RNA polymerase II to the promoters, while the latter stimulates the transfer of the enzyme into the gene body and productive transcription elongation.

The most studied regulators of eukaryotic transcription “pause” are the NELF (Negative Elongation Factor) and DSIF (DRB Sensitivity Inducing Factor) complexes [16,17,18,19]. The DSIF heterodimer, consisting of Spt4 and Spt5 subunits, is a bimodal transcriptional regulator that acts negatively in the “pause” state and positively in the productive elongation stage [20]. At the stage of transcription initiation, the DSIF factor is not able to bind RNA polymerase II since its interaction surface is occupied by the general transcriptional factor TFIIE [21,22]. An association between DSIF and RNA polymerase II is stimulated by the appearance of a newly synthesized RNA [23]. The Spt4 subunit of DSIF modulates RNA polymerase II processivity by restraining the clamp domain motility [22]. The interaction of DSIF with RNA polymerase II stabilizes the binding of the negative transcription factor NELF to the enzyme [24]. Similar to DSIF, the NELF complex directly interacts with a newly synthesized RNA through at least three of its subunits [25,26]. The negative effect of NELF on processivity is partly explained by the fact that it occupies interaction surfaces, which, in the productive RNA polymerase II complex, are associated with positive elongation factors PAF (RNA Polymerase II-Associated Factor) and TFIIS, thus preventing their binding [27]. The promoter state, in which RNA polymerase II is stabilized in the proximal region of the promoter by NELF and DSIF, is called promoter proximal pausing (PrPP) [28,29].

During transcription activation, the NELF and DSIF subunits, as well as Ser2 CTD Rpb1, are phosphorylated by the p-TEFb complex (positive Transcription Elongation Factor), releasing the PrPP state [21]. DSIF becomes a positive regulator of elongation and passes as a part of the RNA polymerase II complex into the gene body, while NELF dissociates and does not participate in further transcription elongation. The place of NELF in the RNA polymerase II elongation complex is occupied by TFIIS, which is necessary for the resumption of RNA synthesis, and the PAF complex, a positive regulator of elongation [27]. To induce PrPP release, p-TEFb can be recruited three different ways: by Brd4/Fs(1)h (Bromodomain protein 4/Female sterile [1] homeotic), as a complex with 7SK RNA, or as a part of the Super Elongation Complex (SEC) [30]. Brd4/Fs(1)h is a bromodomain-containing protein capable of binding to acetylated histones and recruiting the active p-TEFb [31,32]. Although *Drosophila* Brd4/Fs(1) is recruited to chromatin not only through interaction with acetylated histones, treatment of *Drosophila* cells with selective BET inhibitor JQ1 (specific to Brd4/Fs(1)h) leads to a decrease in Brd4/Fs(1)h binding, but not to the background level [33]. Drosophila Brd4/Fs(1)h is represented by two isoforms that have an extended functionality compared with the mammalian protein and are involved not only in the processes occurring on promoters but are also active on insulators and PRE elements [34,35,36].

Promoter proximal pausing is not the only means of post-initiation transcription regulation in Metazoans. Transcription can also be controlled by the poised RNA polymerase II when the enzyme is recruited to the promoter in an inactive state (RNA synthesis is not initiated) [37,38]. There is also a way to control transcription at the step after the action of p-TEFb when Ser2P CTD phosphorylated RNA polymerase II stalls in the promoter region (we call this state the “post-pause” state for brevity). Recently, acute depletion of NELF by auxin degron revealed this distinct “post-pause” stage in mammalian cells [39]. Thus far, there are few ideas regarding which transcription complex can stabilize this state. There is evidence that the PAF complex, a key member of the RNA polymerase II elongation complex involved in the repression of cryptic transcription, plays an active role in the formation of this type of “post-pause” [40,41]. Indeed, both PAF knockdown and rapid degradation have been shown to stimulate the redistribution of RNA polymerase II in gene bodies, in contrast to the degradation of the NELF complex, which only shows a decrease in the amount of RNA polymerase II stalled on promoters [28,41]. Moreover, removal of Spt6, the recruiter of the PAF complex, leads to the same consequences [42]. At present, key aspects of pause mechanisms other than PrPP are only beginning to be described. To illustrate what is currently known, we prepared a scheme of the various types of RNA polymerase II elongation control (Appendix A).

Elongation control of RNA polymerase II is meant to be a way of coordinating various regulatory stimuli [14]. It was observed that developmental genes characterized by a synchronous pattern of transcription activation, such as the *tup*, *sog*, and *eve* patterning genes of early embryogenesis, carry “paused” RNA polymerase II at the promoters [43,44]. The replacement of promoter elements in these genes, which disrupts their ability to “pause”, leads to the heterogeneity of the transcriptional response in the cell population [45,46]. In addition, the temporal synchrony of the transcription activation of these promoters is also disturbed. Apparently, elongation control is used to coordinate not only spatial but also temporal developmental signals. The latter function is the least explored.

The role of the RNA polymerase II “pause” in the coordination of developmental temporal signals was described in most detail by Gaertner and colleagues [47]. In this work, when studying muscle tissue at different stages of *Drosophila* embryogenesis, the presence of RNA polymerase II was found on gene promoters at the developmental stages preceding the stages of their active transcription. Interestingly, the authors of the study found pre-recruited RNA polymerase II not only in the tissue where transcription of these genes would be activated, but also in the other tissue where these genes would remain silent. This suggests a model in which these genes are induced in a tissue-specific manner by factors that release the “pause” of RNA polymerase II. This observation suggests that temporal stimuli (such as hormones) induce RNA polymerase II recruitment in both target and non-target tissues, while spatial stimuli (such as master regulators and morphogens) affect the transcriptional elongation. However, this assumption is preliminary and certainly needs further confirmation.

Recently, we demonstrated that RNA polymerase II and NELF are present on the promoters of ecdysone-induced genes (such as *dhr3*, *hr4*, *eip78c*, etc.), preparing them for the upcoming transcription during *Drosophila* metamorphosis [48]. In this study, we conducted a more global analysis of the role of the RNA polymerase II “pause” during *Drosophila* development by analyzing nonspecific pools of genes that dynamically change their transcription. We obtained a new set of ChIP-Seq data at different stages of *Drosophila* development. These data include ChIP-Seqs of various forms of RNA polymerase II and regulators of transcription elongation NELF-E, Spt5, PAF1 (subunits of NELF, DSIF, and PAF complexes, respectively), and Brd4/Fs(1)h. This allowed us to assess in detail the composition of the “paused” RNA polymerase II complexes formed on promoters at the stages of development preceding the stages of their active work.

## 2. Results

### 2.1. Tissue Collection and Developmental Stage Verification Using Marker Genes Transcription

Investigations focusing on the development of organisms often encounter a problem, namely, difficulties in obtaining material that is highly synchronized in terms of the stage of development. That is why a comparative study of different stages of development in all model organisms is often carried out in early and middle embryogenesis (since embryos are relatively easy to collect). *Drosophila* development has an additional stage, i.e., metamorphosis, which allows one to obtain highly synchronized material. Puparium formation has a clear set of markers at each stage, which makes it possible to obtain specific material. A general idea in investigations of developmental stages is to collect an organism during the most synchronized moment, provide it time to develop, and then collect the material. It must be noted that the more time that passes from the moment the material was collected, the more desynchronized the obtained population becomes.

For the ChIP-Seq analysis, we obtained the material at two time points in embryogenesis (at 2–4 h and at 6–8 h after egg laying (AEL)). We specifically decided to study early and middle embryogenesis in order not to move far from the point of synchronization, i.e., egg laying. In addition, important processes of embryogenesis occur at the following stages: at the 2–4 h stage, active transcription of the genome begins for the first time, and at the 6–8 h stage, specification and determination of cell fate occurs. This is why these embryonic stages are often chosen by researchers [49,50]. We also collected material corresponding to three points of metamorphosis (third instar larvae L3 PS1–3 full gut, white prepupa at 0 h, and 10 h after puparium formation (APF)) (Figure 1). To verify the quality of the material (its correspondence to a certain stage of development), we collected embryos, larvae, and pupae not only at the stages of interest, but also at adjacent time points. At the stage of embryogenesis, we collected embryos from 0 to 12 h AEL (in two-hour increments), and at the stage of metamorphosis, we collected L3 PS1-3 full gut, L3 PS7-9 empty gut, and prepupae at 0–10 h APF (in two-hour increments). To clarify the correspondence of the collected material to the desired stages of development, using RT-PCR, we analyzed the transcription of a number of genes that dynamically change at these stages and compared the change in their levels with the earlier obtained data (the expected transcription peak for each gene, based on previously published data, is indicated by an asterisk) (Figure 1) [51]. For the analysis of the embryos, we chose patterning genes *giant*, *snail*, *thisbe*, and *twist*. For the analysis of the larvae and prepupae, we selected genes activated by ecdysone: early *sgs3* and *br* and early–late *dhr3* and *hr4*. The pattern of changes in the gene transcription in the analyzed tissues corresponded to the expected pattern (previously published) [51]. We concluded that the material we collected was suitable for the planned ChIP-Seq analysis.

### 2.2. Genes Induced at 6–8 h of Embryogenesis Moderately Use the “Paused” RNA Polymerase II to Prepare for the Upcoming Transcription

To double-check the previously obtained information regarding the role of the RNA polymerase II “pause” in preparing the promoters for the upcoming transcription in embryogenesis, we started from the analysis of various pools of genes activated at this *Drosophila* developmental stage [47].

First, for the analysis, we selected a pool of genes whose transcription is induced at the 6–8 h AEL (after eggs laying) stage by at least five times relative to the 2–4 h AEL stage (“6–8 h genes”). The total number of selected genes was 420 (which corresponded to 1253 transcripts). The selection of this set of genes was performed using data previously published by Graveley and colleagues (using data provided in Supplementary Table 9 of their manuscript representing the FPKM levels in the genes of various Drosophila developmental stages, total number of genes was 15,139) [51]. We tested the quality of our selection by calculating the average RNA-seq signal level using previously published raw data (Figure 2). The obtained panel clearly demonstrated that the selected gene pool was inactive at 2–4 h AEL but was activated at 6–8 h AEL. The list of selected regions corresponding to “6–8 h genes” is provided as Appendix A.

We obtained the averaged binding profiles of RNA polymerase II isoforms and factors that control the elongation on the resulting set of “6–8 h genes” (Figure 2). Our ChIP-Seq analysis demonstrated active recruitment of RNA polymerase II to the promoters of the “6–8 h genes” at the 6–8 h AEL stage and its phosphorylation. The regulators of transcription elongation NELF, DSIF, and PAF were also recruited to these promoters at the 6–8 h AEL stage. Contrary behavior at this developmental stage was demonstrated by Brd4/Fs(1)h, which we detected on the promoters of the “6–8 h genes” at the previous developmental stage of 2–4 h AEL, and which left the promoters in transition to the stage of active transcription of this gene pool. We did not detect the presence of DSIF or PAF in the gene bodies of the “6–8 h genes” at the stage of their active transcription at 6–8 h AEL, although they were all expected to be there. We attribute the odd behavior of factors controlling elongation in the “6–8 h genes” to the atypical composition of the “paused” RNA polymerase II complex (lack of Brd4/Fs(1)h).

We observed the presence of some RNA polymerase II and elongation regulators at the 2–4 h AEL stage preceding the transcriptional activation stage of the 6–8 h AEL gene pool. This was especially evident from the substantial enrichment of the promoters by NELF-E at the 2–4 h AEL stage. However, it did not appear that the genes induced at 6–8 h AEL actively used the “paused” RNA polymerase II to prepare the promoters for the upcoming transcription. It is also possible that the level of this “paused” RNA polymerase II was rather low and it could not be easily detected on averaged profiles. The level of RNA polymerase II on these promoters increased significantly during the transition from 2–4 h to 6–8 h AEL.

To estimate the exact proportion of “6–8 h gene” promoters bound by RNA polymerase II at the 2–4 h stage, we performed a cluster analysis according to the amount of Rpb3 (total RNA polymerase II). This analysis demonstrated that 20.8% of the “6–8 h gene” promoters (corresponding to Cluster 1) possess RNA polymerase II before their activation at the 6–8 h AEL stage (Appendix A). It important to add that all the analyzed factors, such as NELF-E, Spt5, PAF1, and Brd4/Fs(1)h, were also present in Cluster 1 with “paused” RNA polymerase II, but not in Cluster 2 that lacked the enzyme.

### 2.3. Genes Induced at 10–12 h AEL, but Not Genes Induced at 14–16 h AEL, Possess “Paused” Pol II on the Promoters at 6–8 h AEL

Next, we decided to test whether genes from the later stages of embryogenesis used the RNA polymerase II “pause” to prepare for the upcoming transcription. We selected genes whose transcription was induced at least five times at the 10–12 h AEL stage compared with the 6–8 h stage level (“10–12 h genes”) and another set of genes that were induced at the 14–16 h AEL stage at least five times compared with the 10–12 h AEL genes (“14–16 h genes”). The total number of selected “10–12 h genes” was 462 (which corresponded to 1188 transcripts) and 548 for the “14–16 h genes” set (which corresponded to 1379 transcripts). The quality of the selection was tested in the same way as was carried out for the pool of “6–8 h genes” by obtaining the averaged profiles of the RNA-Seq signal (Figure 3). The lists of selected regions corresponding to “10–12 h genes” and “14–16 h genes” are provided as Appendix A, respectively.

We found the presence of a substantial amount of RNA polymerase II on the promoters of “10–12 h genes” at the 6–8 h AEL developmental stage (Figure 3). RNA polymerase II was not detected on “10–12 h genes” at the 2–4 h AEL stage. That is, the “10–12 h genes” use the RNA polymerase II “pause” to prepare the promoter for transcription, but do not recruit it there much in advance. We detected a significant amount of modified RNA polymerase II at the developmental stage of 6–8 h AEL on “10–12 h genes”, both phosphorylated at Ser5 and Ser2 of Rpb1 CTD. This indicates that at least partially these genes use the “post-pause” to prepare their promoters for upcoming transcription. In support of this, we detected the presence of NELF, DSIF, and PAF complexes together with RNA polymerase II at the 6–8 h AEL stage on “10–12 h genes” (Figure 4). Brd4/Fs(1)h was absent on the promoters of “10–12 h genes” at the 6–8 h AEL stage.

To estimate the exact proportion of the “10–12 h genes” promoters bound by RNA polymerase II at the 6–8 h stage, we performed a cluster analysis according to the amount of Rpb3 (total RNA polymerase II) (Appendix A). This analysis demonstrated that 12.7% of the “10–12 h gene” promoters (corresponding to Cluster 1) possessed RNA polymerase II before their activation at the 10–12 h AEL stage (Appendix A). We found NELF-E, Spt5, and PAF1 present in Cluster 1 with “paused” RNA polymerase II, but not in Cluster 2 that lacked the enzyme. Brd4/Fs(1)h was absent from the promoters of the entire set of the “10–12 h genes”. We explain the relatively small percentage of promoters associated with RNA polymerase II by the limitations of the ChIP-Seq: the whole-embryo analysis did not make it possible to describe the state of the genes that are activated in a highly tissue-specific manner.

The generation of averaged profiles in the region close to TSS revealed the presence of an additional peak of RNA polymerase II, phosphorylated for both Pol II CTD Ser2 and Pol II Ser5, within the distal part of the promoters, which cannot be correlated with the accumulation of any analyzed elongation control factors (Appendix A).

To determine how much in advance, before transcription, developmental genes recruit RNA polymerase II to their promoters, we analyzed the pool of “14–16 h genes” in our datasets (lower panels of Figure 3 and Figure 4). We did not detect a substantial binding level of RNA polymerase II and elongation regulators for the “14–16 h gene” promoters at the 6–8 h AEL. This result suggests that, in mid-embryogenesis, gene promoters use RNA polymerase II pre-recruitment to prepare the promoters for transcription at later stages, but not much in advance. It is worth noting that, at the 6–8 h AEL stage, we detected a small amount of RNA polymerase II on genes activated at the 14–16 h AEL stage. This small amount of RNA polymerase II was present mostly not on the promoters but in the genes’ bodies. That is, on the promoters of this gene pool at the 6–8 h AEL stage, there was practically no control of the transition of RNA polymerase II into the genes’ bodies (there was no control of transcription elongation).

### 2.4. Brd4/Fs(1)h Protein Is Maternity Loaded and Its Expression Is Delayed until the Mid-Embryogenesis

At this point in the study, we already analyzed three different pools of genes (induced at 6–8 h, 10–12 h, and 14–16 h AEL). In all the analyzed pools, we observed a low level of Brd4/Fs(1)h binding to promoters in embryogenesis (with the exception of the presence of some Brd4/Fs(1)h on the promoters of the “6–8 h genes” at the 2–4 h AEL stage). The generation of the Brd4/Fs(1)h binding profile for the average *Drosophila* gene in ChIP-Seqs at 2–4 h and 6–8 h provided a similar result: the level of Brd4/Fs(1)h binding decreased when moving from 2–4 h to 6–8 h AEL (Appendix A). We were interested in the reason for the lack of Brd4/Fs(1)h binding to promoters at the developmental stage of 6–8 h AEL. We suggested that this can be caused either by the absence of the protein itself or by the inability of the protein to bind chromatin at this developmental stage. To differentiate between these two possibilities, we decided to investigate Brd4/Fs(1)h expression in early and mid-embryogenesis. To perform this, we analyzed staged embryos by Western blotting using antibodies against Brd4/Fs(1)h (Appendix A). We found that, in a protein extract, Brd4/Fs(1)h was present only in embryos at 0–1 h AEL and it disappeared at the later stages. Its protein level began to recover only from 5 to 6 h AEL. A possible explanation is that Brd4/Fs(1)h is deposited as a protein in the oocyte (its high level at the 0–1 h stage cannot be explained by the activity of the gene since the genome is not yet active at this stage). However, the zygotic expression of Brd4/Fs(1)h does not occur during the first wave of zygotic genome activation (ZGA) and is delayed until mid-embryogenesis. This is why we observed a certain level of Brd4/Fs(1)h binding at the 2–4 h stage (the chromatin-associated protein probably disappeared a little later than the protein from the nucleosol) and a decrease in the level of Brd4/Fs(1)h binding in the genome at the 6–8 h AEL stage. Our results show that there is an interesting stage in *Drosophila* development when the Brd4/Fs(1)h regulator of elongation is not involved in gene transcription.

### 2.5. Genes Induced during Metamorphosis Use Promoter-Proximal Pausing to Prepare Promoters for Upcoming Transcription

To explore whether the RNA polymerase II “pause” is used to prepare promoters for upcoming transcription at other developmental stages, we took advantage of *Drosophila* metamorphosis and performed ChIP-Seqs on homogeneously staged L3 PS1-3 full gut, white prepupa, and prepupa 10 h after puparium formation (APF). For analysis, we selected two pools of genes: genes induced in wandering larvae during the puparium formation, i.e., “WL genes” (wandering larvae (WL)), and genes activated 24 h after the formation of the white prepupa, i.e., “WPP +24 h genes” (white prepupa (WPP)). The total number of “WL genes” that increased their transcriptional level by at least five times at the L3 PS7-9 empty gut stage compared with the L3 PS 1-3 full gut stage was 303 (which corresponds to 852 transcripts). The total number of “WPP +24 h genes” that increased their transcriptional level by at least five times at the WPP +24 h stage compared with WPP + 12 h stage was 62 (which corresponds to 108 transcripts). The lists of selected regions corresponding to “WL genes” and “WPP +24 h genes” are provided as Appendix A, respectively.

First, we analyzed the state of RNA polymerase II at the promoters of the “WL genes” at different stages of metamorphosis (Figure 5, top panels). The “WL genes” demonstrated binding of RNA polymerase II to their promoters at the L3 PS1-3 full gut stage. The RNA polymerase II level did not increase during the transition to the white prepupa stage. The degree of RNA polymerase II phosphorylation at Ser5 Rpb1 CTD also did not change. However, we observed a significant increase in the level of Ser2 Rpb1 CTD phosphorylation, which was probably the result of increased p-TEFb activity. That is, the “WL genes” mainly used promoter-proximal pausing to prepare for the upcoming transcription. The ChIP-Seq results of the elongation regulators support this conclusion (Figure 6). The promoter-bound level of NELF and DSIF did not increase in the promoters of the “WL genes” during the transition from the L3 PS1-3 full gut stage to the wandering stage. We observed a significant increase in the binding level for the PAF complex. Interestingly, we detected a significant level of Brd4/Fs(1)h on the promoters of genes active during metamorphosis. Apparently, being expressed in mid-embryogenesis, this protein actively participates in the transcriptional regulation. Brd4/Fs(1)h is involved both in the preparation of the promoter for transcription and in the active transcription of “WL genes”.

With active transcription of the “WL genes”, we observed a significant increase in binding of the DSIF and PAF complexes and Brd4/Fs(1)h in the gene bodies. Certainly, these regulators are closely associated with RNA polymerase II not only in the promoter but also during its movement in the gene bodies. This result is in complete agreement with the previously described behavior of these complexes during active transcription [41,52,53,54].

To estimate the exact proportion of the “WL gene” promoters bound by RNA polymerase II at the L3 PS1-3 full gut stage, we performed a cluster analysis according to the amount of Rpb3 (total RNA polymerase II) (Appendix A). This analysis demonstrated that 26% of the “WL gene” promoters (corresponding to Cluster 1) possessed RNA polymerase II before their activation at the L3 PS7-9 empty gut stage (Appendix A). We found NELF-E, Spt5, PAF1, and Brd4/Fs(1)h present in Cluster 1 with “paused” RNA polymerase II, but not in Cluster 2 that lacked the enzyme. The generation of averaged profiles in the region close to TSS revealed the presence of an additional peak of RNA polymerase II, phosphorylated for both Pol II CTD Ser2 and Pol II Ser5, within the proximal part of the promoter, which cannot be correlated with the accumulation of any analyzed elongation control factors (Appendix A).

As in the study of embryogenesis, we decided to look at whether RNA polymerase II is recruited to genes that are activated later in development than “WL genes”. We analyzed the obtained ChIP-Seqs on the pool of genes activated 24 h after puparium formation, i.e., “WPP +24 h” (bottom panels in Figure 5 and Figure 6). We did not detect the presence of a significant amount of RNA polymerase II or elongation regulators on the promoters of this pool of genes. Thus, as in the course of embryogenesis, during metamorphosis, *Drosophila* uses the RNA polymerase II “pause” to prepare promoters for transcription at previous stages of development, but not too much in advance.

### 2.6. The Pool of Ecdysone-Dependent Genes That Is Iteratively Induced during Development Is Controlled by Promoter-Proximal Pausing

In the results presented above, we analyzed pools of genes selected according to the common rules (so that the gene was activated at least five times compared with the previous developmental stage). To control our selection scheme, we decided to analyze a pool of genes selected according to a completely different principle. We chose a pool of genes that were repeatedly activated in development, being induced by the 20-hydroxyecdysone hormone (ecdysone). Since we obtained the ChIP-Seqs for the material of whole embryos, larvae, and prepupae (but not for tissues), we chose a pool of ecdysone-inducible genes from the study characterizing the ecdysone response in many cell lines at once (total number of genes in this set was 68 which corresponded to 236 transcripts) [55]. The list of regions corresponding to stably induced ecdysone-dependent genes is provided as Appendix A.

First, we checked at what developmental stages a selected pool of ecdysone-inducible genes is activated (Figure 7). As expected, we found active transcription of this pool at 10–12 h of embryogenesis and during puparium formation. The overall changes in the level of RNA polymerase II and elongation control factors binding on ecdysone-inducible genes were consistent with what we observed for other genes activated at the 10–12 h AEL and wandering larva stages. Despite the low level of active transcription at the 6–8 h AEL stage of embryogenesis, we detected active recruitment to promoters of RNA polymerase II genes and its phosphorylation at Ser5 Rpb1 CTD and even Ser2 Rpb1 CTD (Figure 7). We also observed recruitment of NELF-E, Spt5, and PAF1 to ecdysone-inducible promoters at 6–8 h of embryogenesis (Figure 8).

During metamorphosis, the pool of ecdysone-inducible genes was found to be regulated in the same way as the general pool of genes activated at the WL stage. We observed the presence of RNA polymerase II on the promoters of ecdysone-inducible genes up to the stage of their active transcription, and its level did not increase upon activation. We detected an increase only in the degree of RNA polymerase II phosphorylation at Ser2 Rpb1 CTD (Figure 7). That is, the transcription of the ecdysone-inducible gene pool during metamorphosis was controlled by promoter-proximal pausing. We detected the presence of all elongation control factors on the promoters of ecdysone-inducible genes both before and during the state of their active transcription during puparium formation (Figure 8).

## 3. Discussion

This study aimed to fill a gap in the knowledge regarding how *Drosophila* uses the RNA polymerase II “pause” to prepare promoters for active transcription at the next stage of development. The main purpose was to determine whether the “pause” is involved in the preparation of genes for transcription at various stages of development. *Drosophila* development provides a very convenient opportunity for this by allowing us to obtain material that is highly synchronized in terms of developmental stages, not only during embryogenesis but also during the metamorphosis phase.

### 3.1. Genes That become Active in Mid-Embryogenesis and in the Early Stages of Metamorphosis Use Elongation Control to Prepare Promoters for Transcription

Analyzing the pools of “6–8 h genes”, “10–12 h genes”, and “WL genes” activating during mid-embryogenesis and metamorphosis, we observed RNA polymerase II binding to promoters at the stages preceding the stages of their active transcription (Figure 9). The composition and properties of the “paused” RNA polymerase II complexes were found to differ in mid-embryogenesis and metamorphosis. The “pause” of RNA polymerase II in mid-embryogenesis is characterized by phosphorylation of its Pol II CTD not only by Ser5 but also by Ser2, which corresponds to the “post-pause” state, operating at the transcriptional step after the activity of the p-TEFb complex [39,41]. In the course of metamorphosis, the genes use the more well-described type of RNA polymerase II “pause”, i.e., promoter-proximal pausing, which is characterized by a high level of Pol II CTD Ser5 phosphorylation and a low degree of Pol II CTD Ser2 phosphorylation [29]. The composition of the “paused” RNA polymerase II complexes in embryogenesis and metamorphosis differs in the number of associated elongators; the embryonic “pause” complex lacks Brd4/Fs(1)h due to the low expression level of this protein at this stage of development. The rest of the studied elongation regulators, namely, NELF, DSIF, and PAF, were found to be involved in the RNA polymerase II “pause” both in embryogenesis and metamorphosis.

Everything described above regarding the RNA polymerase II “pause” both in embryogenesis and metamorphosis is related to the RNA polymerase II peak formed in the proximal part of the promoters. However, a close inspection of the distribution of RNA polymerase II across TSS regions revealed that the “paused” promoters, both at 6–8 h of embryogenesis and at the L3 PS 1-3 full gut stage of metamorphosis, had an additional downstream peak of RNA polymerase II and its isoforms Pol II CTD Ser5P and Pol II CTD Ser2P (Appendix A). We found no presence of NELF, DSIF, PAF, or Brd4/Fs(1)h in this distal peak. A possible force that may hold the RNA polymerase II in this position is the structure of chromatin, as was previously suggested [39]. However, we believe that some additional elongation control factors that have yet to be found may participate in the formation of this distal “pause” peak. We would like the readers of our article to consider these ideas as preliminary as the performed ChIP-Seq assays does not have sufficient sensitivity to resolve such close peaks of RNA polymerase II at the promoters (and for these purposes, it is better to perform ChIP-exo, or even ChIP-nexus, assays) [56].

The performed cluster analysis showed that most of the promoters were not associated with RNA polymerase II before their activation, and our conclusions are valid only for some of the genes preparing for transcription. We attribute this to the limitations of ChIP-Seq. Because we analyzed the entire embryo and larva, we were not able to detect tissue-specific binding events. Single-cell techniques may help to overcome this problem, and the implementation of such techniques appears to be a good development for the current work.

### 3.2. Elongation Regulators Are Recruited to Promoters during Drosophila Development Only When RNA Polymerase II Is Recruited

It would seem that this is a natural conclusion, since elongation regulators directly interact with RNA polymerase II, but this is not very obvious. The process of elongation regulators recruitment to RNA polymerase II is very unclear and it may well be a multistage process. Additionally, the step of this multistage process may well be the recruitment of elongation regulators onto chromatin through interaction with DNA-binding proteins, and not directly with RNA polymerase II. With this recruitment mechanism, even in the absence of RNA polymerase II, we would detect binding of some elongation regulators to promoters due to their recruitment by DNA-binding proteins. However, in all the analyzed pools of genes that did not contain RNA polymerase II on the promoters (the pools of 14–16 h of embryogenesis and WPP +24 h), we did not observe the binding of elongation regulators with the promoters. Moreover, the cluster analysis showed that all the analyzed elongation regulators fell into the cluster of genes containing RNA polymerase on the promoters. That is, at least at the stages of *Drosophila* development that we analyzed, the recruitment of elongation regulators to the promoters occurred together with the recruitment of RNA polymerase II.

It is worth noting that, in previous studies, we detected the binding of elongation regulators with DNA in the absence of RNA polymerase II. In our recent works, we described that the NELF-A subunit of the NELF complex is able to bind not only promoters but also enhancers and PRE elements containing a relatively low level of RNA polymerase II [34,48]. The distribution profile of NELF-A in the genome indicates that this particular NELF subunit can be recruited by DNA-binding proteins separately from other subunits of this complex and, most importantly, separately from RNA polymerase II. Additionally, the recruitment of this subunit may well be an early stage in the assembly of the full NELF complex.

The *Drosophila* Brd4/Fs(1)h protein was previously found to be present not only in promoters and enhancers but also in sites enriched in architectural proteins, mostly not associated with RNA polymerase II [35]. That is, Brd4/Fs(1)h recruitment can also occur not directly to RNA polymerase II, but through an intermediate step of its recruitment to chromatin via DNA-binding (architectural) proteins.

It seems that some elongation regulators can indeed be recruited by DNA-binding proteins as a preliminary step in their binding to RNA polymerase II; however, judging by the data of this article, this does not occur on promoters.

### 3.3. Release of Developmental Genes from Elongation Control

Our data suggest that, during *Drosophila* development, genes prepare in advance for the upcoming transcription by pausing the RNA polymerase II at their promoters. It is assumed that productive transcription of these genes at the appropriate stage is achieved by resolving this “pause”. In the case of promoter-proximal pausing, this is the recruitment of the p-TEFb complex to promoters or its activation if it is pre-recruited in an inactive HEXIM-suppressed state [57]. In the case of a “post-pause”, the “pause” release can be induced by the recruitment or modification of a certain subunit of the PAF complex, although it is too early to discuss the exact mechanism for this type of pause [41].

A not entirely clear but interesting question concerns how the increase in the concentration of “pause-releasing” complexes on the targeted promoters is achieved. Is it gene-specific, as in the case with heat shock genes activated by recruitment of HSF1, which stimulates elongation [15]? Or can there be a global change in the intracellular concentration of complexes stimulating elongation at certain stages of development? The change in the expression level of Brd4/Fs(1)h during development that we observed indicates that the second possibility may well be implemented. It is quite probable that for some genes that form partially prepared RNA polymerase II complexes on promoters, an increase in the concentration of Brd4/Fs(1)h in mid-embryogenesis can stimulate their productive transcription.

The most advanced works in this area, namely, the control of gene transcription through the intracellular level of coregulators, refers to genes controlled by poised Pol II and released by TFIIH complex. Some time ago, it was demonstrated that a change in the concentration of TFIIH (a general transcriptional factor stimulating DNA melting and transcription initiation, that is, exit from the poised Pol II state) is controlled by the level of glucose [58]. More recently, the intracellular level of TFIIH has been linked to the transcription of genes responsible for proliferative cell potential using a single-cell approach [59]. It would be extremely interesting to study the level of other regulators that stimulate the release of various types of RNA polymerase “pauses” in cells during development as well as in the case of any external stimuli or the progression of pathologies.

## 4. Conclusions

The transcription of developmental genes is under the control of a variety of regulatory systems that control the timing and specificity of transcription in a particular tissue as well as under the influence of master regulator proteins that control transcription in a particular part of the body. It takes time to implement and coordinate all these stimuli. Not surprisingly, developmental genes control their transcription by controlling productive elongation. This approach helps to form a transcriptional hub in the promoter region and ensures the specificity of all the necessary interactions with RNA polymerase II and GTFs. The study of the dynamics of such hubs in development can help us to better understand the mechanisms of transcription regulation in general. *Drosophila*’s rapid development is a convenient experimental model for this goal.

## 5. Materials and Methods

### 5.1. Collection of the Material Corresponding to Different Drosophila Developmental Stages

The flies of Oregon-R-modENCODE stock were used (corresponds to Bloomington stock #25211). Embryos were collected in the fly cages for 2 h using apple juice agar plates. Then plates were incubated at the 25 °C for the required period of time. The time after egg laying (AEL) was calculated starting from the moment the agar plates were placed in the fly cages. To reduce an effect of the retention of embryos inside flies (which occurs in aging *Drosophila*), only flies aged 3–5 days after eclosion were used to collect embryos.

L3 larvae, corresponding to the puff stages 1–3 full gut and puff stages 7–9 empty gut, were collected by culturing larvae in the fly media supplemented with 0.05% bromophenol blue to mark the guts of feeding animals. Prepupae corresponding to 0–1 h after puparium formation (APF) were collected according to description of this developmental stage: white motionless prepupae starting to evert their anterior spiracles. Prepupae corresponding to later developmental stages were collected at the stage of white prepupa and then incubated at the 25 °C for the required period of time.

For the ChIP-Seq experiments embryos were dechorionized, washed and homogenized in a buffer (60 mM KCl, 15 mM NaCl, 4 mM MgCl_2_, 15 mM HEPES pH7.6, 0,5% Triton X-100) containing 0.7% of formaldehyde for 10 min and incubated for 5 min with 125 mM Glycine. Then cells were washed for three times with homogenization buffer. The remaining ChIP protocol was performed as described previously [60,61].

To validate our material collected from the various developmental stages we measured transcription of various genes (these data are discussed in a first section of the Results and provided in Figure 1).

### 5.2. ChIP-Seq Analysis

The chromatin immunoprecipitation (ChIP) was performed and analyzed exactly as previously described [42,61]. ChIP-Seq libraries were obtained using the NEBNext Ultra^TM^ II DNA library preparation kit (New England Biolabs, Ipswich, USA). Only the library fragments of 350–500 bp were subjected to NGS sequencing. Next generation sequencing was performed by Evrogen (evrogen.ru) with the Illumina NovaSeq6000 sequencer. For each of the ChIP-Seq libraries approximately 3–8 millions of unique paired-end reads were obtained. The paired-end reads in FastQ format were mapped to the *Drosophila* genome assembly dm6 using HISAT2 [62] and filtered (with minimum MAPQ quality score = 5). Deeptool2 package (Freiburg, Germany) was used for the further analysis of the obtained data [63]. BigWig files were generated using bamCoverage 3.0.2 with scores representing number of reads normalized by the size of the library (the protein binding levels were normalized to the genome content, calculated as RPGC: number of reads per bin/(total number of mapped reads * fragment length/effective genome size)) [63]. The final BigWig files (representing the protein binding profiles) were obtained using BigWigCompare tool as ratio of ChIP signal to input (all inputs were preliminary smoothed over a 1 kb window). Pile-up profiles were calculated as a median level of protein binding (except for RNA-Seqs profiles and ChIP-Seqs profiles analyzed at the ecdysone-induced set of genes, which were calculated as median levels). Clustering analysis was performed using plotHeatmap tool of deepTool2 package using ChIP-Seq Rpb3 (which corresponds to the total Pol II) [63]. The ChIP-Seq data were clustered using the values at the TSS. K-means clustering algorithm was selected and the number of clusters to compute was set to 2 (which resulted in Cluster 1, enriched with Rpb3, and Cluster 2, depleted in Rpb3).

Sets of genes induced at the specific stages of *Drosophila* development were selected using information from Supplementary Table 9 of previously published study (this table represents FPKM levels at FlyBase 5.12 Genes of various developmental stages) [51]. For example, the set of genes induced at 6–8 h AEL were selected following these rules: to demonstrate (1) more than five FPKM level at the 6–8 h AEL stage and (2) more than five times increase in their transcription level in relation to 2–4 h AEL stage. For the selection of genes induced at other stages of development, the same rules were applied (number of selected genes and selection rules are described in the Results). All selected pools of genes analyzed in the manuscript are provided as Supplementary tables in a “bed” format.

The set of genes induced in wandering larvae “WL genes” was selected by comparing transcriptional levels of genes at L3 PS7-9 empty gut and L3 PS1-3 full gut stages. We had to use L3 PS7-9 empty gut instead of white prepupa 0 h stage as the latter was absent in analysis performed by Graveley et al. We consider this substitution to be insignificant since the L3 PS7-9 empty gut and the white prepupa 0 h stages are separated by only a few hours and represent a common transcriptional induction wave that is primarily a response to ecdysone. Our data on the distribution of various forms of RNA polymerase II on this gene set confirms this conclusion, we saw a substantial increase in the phosphorylation level of Ser2 Rpb1 CTD at these genes during the transition between the L3 PS1-3 full gut and white prepupa stages, indicating the active work of this gene set at the stage of white prepupa.

To verify our selection strategy, we provided averaged profiles of RNA-Seqs at the selected pools of genes in each figure. We used raw RNA-Seq data obtained from ModENCODE web-server, which were previously used by Graveley and colleagues for the analysis of *Drosophila* developmental transcriptome [51,64]. We did not use any figures or text from the previously published manuscripts, only data deposited in free access databases.

The Galaxy-P platform (Freiburg, Germany) was used for analysis of ChIP-Seq data [65]. All obtained ChIP-Seq data were deposited into the Gene Expression Omnibus—GSE210971. 

### 5.3. Antibodies and Western Blotting

Rabbit polyclonal antibodies against Rpb3 (1–275 aa), Spt5 (917–1117 aa), NELF E (2–279 aa), PAF1 (1–234 aa), Fs(1)h/Brd4 (120–386 aa) were obtained and described previously in our lab [48,66]. All antibodies were affinity purified. All of them were tested in ChIP experiments using *hsp70* gene induced by heat shock [53]. Antibodies production was performed according to procedures outlined in the NIH (USA) Guide for the Care and Use of Laboratory Animals. The protocol used was approved by the Committee on Bioethics of the Institute of Gene Biology of the Russian Academy of Sciences. All procedures were performed under conditions designed to minimize suffering.

Antibodies against RNA polymerase II CTD repeat (phospho S2) (Ab5095), and RNA polymerase II CTD repeat (phospho S5) (Ab5131) were purchased in Abcam (Cambridge, United Kingdom).

To obtain protein extracts for Western blotting of various embryogenesis stages, embryos were collected using apple juice agar plates in the same way as was performed for the collection of ChIP-Seq material (the only difference is 1h-increment between different time points). For Western blotting embryos were dechorionized, washed, and homogenized by triple sonication in a RIPA buffer (50 mM HEPES-KOH, pH 7.9; 140 mM NaCl, 1% Triton X-100, 0.1% Na deoxycholate, and 0.1% SDS). Protein extracts were treated with DNAse I at 37 °C for 15 min and centrifuged at 16,000× *g* for 20 min. Then supplemented with SDS loading buffer containing DTT and loaded on a polyacrylamide gel.

## Figures and Tables

**Figure 1 ijms-23-10662-f001:**
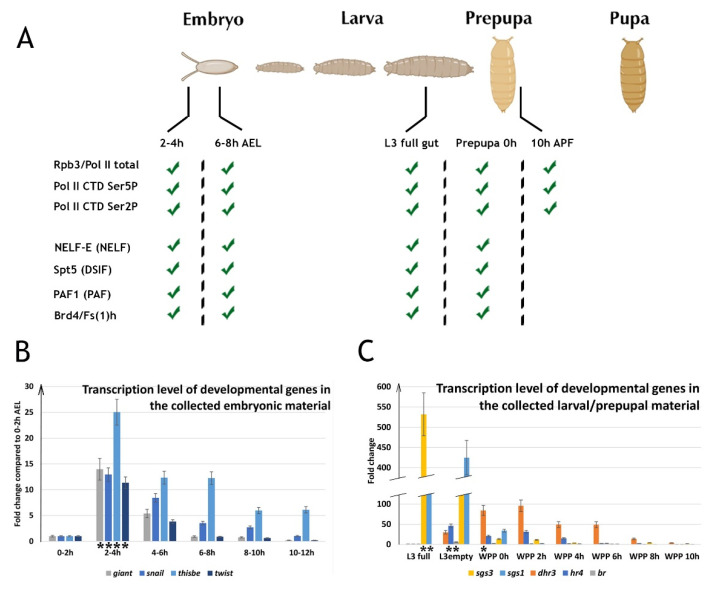
Validation and characterization of the material collected at different developmental stages of *Drosophila*. (**A**) Scheme describing the ChIP-Seqs dataset obtained in the current study. For ChIP-Seq analysis, we collected embryos, larvae, and prepupa at different stages of *Drosophila* development. We obtained the material at two time points in embryogenesis (2–4 h and 6–8 h after egg laying (AEL)) and at three points of metamorphosis (L3 PS1-3 full gut, white prepupa (WPP) at 0 h, and 10 h after puparium formation (APF)). To verify the quality of the material collected at the target and adjacent time points of embryogenesis (**B**) and metamorphosis (**C**) (its correspondence to a certain stage of development) transcriptional levels of developmental genes were assessed by qRT-PCR. The patterning genes *giant*, *snail*, *thisbe*, and *twist* were chosen for the analysis of the embryonic material. The ecdysone-responsive genes: early *sgs1*, *sgs3,* and *br* and early–late *dhr3* and *hr4*, were chosen for the analysis of the larvae and prepupae. The *Y*-axis units represent relative transcriptional level: in relation to 0–2 h AEL stage for the embryonic material and in relation to L3 PS1-3 full gut for the larvae and prepupae (except for *sgs1* and *sgs3* transcription level which is presented in relation to WPP 6 h stage). The data are mean values from three technical experimental repeats. The expected transcription peak for each gene, based on previously published data, is indicated by an asterisk (data were taken from *flybase.org* (accessed on 10 August 2022)) [51].

**Figure 2 ijms-23-10662-f002:**
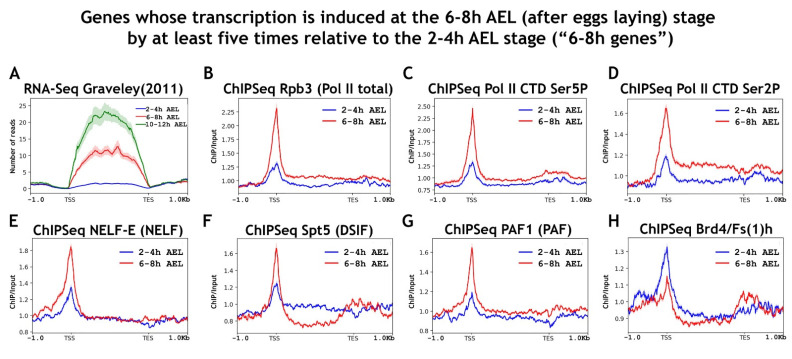
Average binding profiles of various RNA polymerase II isoforms and elongation regulators NELF, DSIF, PAF, and Brd4/Fs(1)h on genes induced at the 6–8 h AEL (after egg laying) stage of *Drosophila* embryogenesis. Average distribution of (**A**) RNA-seq signal and ChIP-Seq signal of (**B**) Rpb3 (Pol II subunit), (**C**) Pol II Ser5P CTD, (**D**) Pol II Ser2P CTD, (**E**) NELF-E, (**F**) Spt5, (**G**) PAF1, and (**H**) Brd4/Fs(1)h across the genes induced at the 6–8 h stage of *Drosophila* embryogenesis by at least five times relative to the 2–4 h AEL stage (total number genes in “6–8 h genes” set are 420 (corresponded to 1253 transcripts)). ChIP-Seqs were performed on whole embryos of 2–4 h (blue line) and 6–8 h (red line) after eggs laying (AEL). A pool of “6–8 h AEL genes” was selected using previously published data [51]. Protein binding levels were calculated as an enrichment (ratio of the corresponding ChIP-Seq signal to the input DNA). Average profiles were generated using the metagene mode (introns were ignored and gene bodies were scaled to 2 kb) and calculated as the median of the protein binding signal. Average profiles of the RNA-seq signal were calculated as the mean level. The standard error appears on the graphs as a lighter area around the main line of the profiles. Abbreviations: TSS: transcription start site; TES: transcription end site.

**Figure 3 ijms-23-10662-f003:**
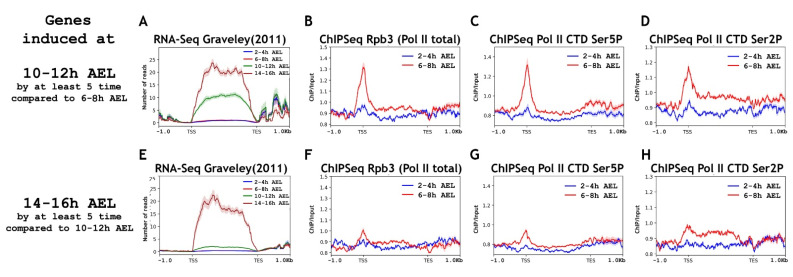
Average binding profiles of various Pol II isoforms on genes induced at the 10–12 h and 14–16 h AEL (after eggs laying) stages of *Drosophila* embryogenesis. Average of the (**A**) RNA-seq signal and an average distribution of (**B**) Rpb3 (Pol II subunit), (**C**) Pol II Ser5P CTD, and (**D**) Pol II Ser2P CTD binding across the genes induced at 10–12 h AEL (after egg laying) of *Drosophila* embryogenesis (total 462 genes/1188 transcripts). Average of the (**E**) RNA-seq signal and an average distribution of (**F**) Rpb3 (Pol II subunit), (**G**) Pol II Ser5P CTD, and (**H**) Pol II Ser2P CTD binding across the genes induced at 14–16 h AEL (after egg laying) of *Drosophila* embryogenesis (total 548 genes/1379 transcripts). ChIP-Seqs were performed on whole embryos of 2–4 h (blue line) and 6–8 h (red line) after eggs laying (AEL). Pools of “10–12 h AEL genes” and “14–16 h AEL genes” were selected according to the common rule, to be induced at least five times compared with “6–8 h AEL” and “10–12 h AEL” stages, respectively. Protein binding levels were calculated as an enrichment (ratio of the corresponding ChIP-Seq signal to the input DNA). Average profiles were generated using the metagene mode (introns were ignored and gene bodies were scaled to 2 kb) and calculated as the median of the protein binding signal. Average profiles of the RNA-seq signal were calculated as the mean level. The standard error appears on the graphs as a lighter area around the main line of the profiles. Abbreviations: TSS: transcription start site; TES: transcription end site.

**Figure 4 ijms-23-10662-f004:**
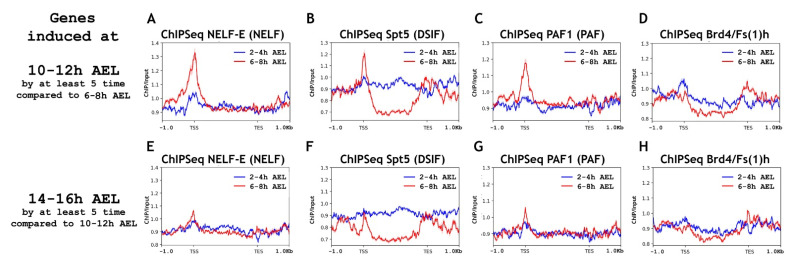
Average binding profiles of NELF-E (NELF), Spt5 (DSIF), PAF1 (PAF), and Bdr4/Fs(1)h on genes induced at the 10–12 h and 14–16 h AEL (after egg laying) stages of *Drosophila* embryogenesis. Average distribution of (**A**) NELF-E (NELF complex), (**B**) Spt5 (DSIF complex), (**C**) PAF1 (PAF complex), and (**D**) Brd4/Fs(1)h binding across the genes induced at 10–12 h AEL (after egg laying) of *Drosophila* embryogenesis (total 462 genes/1188 transcripts). Average distribution of (**E**) NELF-E (NELF complex), (**F**) Spt5 (DSIF complex), (**G**) PAF1 (PAF complex), and (**H**) Brd4/Fs(1)h binding across the genes induced at 14–16 h AEL (after egg laying) of *Drosophila* embryogenesis (total 548 genes/1379 transcripts). ChIP-Seqs were performed on whole embryos of 2–4 h (blue line) and 6–8 h (red line) after eggs laying (AEL). Pools of “10–12 h AEL genes” and “14–16 h AEL genes” were selected according to the common rule, to be induced at least five times compared with “6–8 h AEL” and “10–12 h AEL” stages, respectively. Protein binding levels were calculated as an enrichment (ratio of the corresponding ChIP-Seq signal to the input DNA). Average profiles were generated using the metagene mode (introns were ignored and gene bodies were scaled to 2 kb) and calculated as the median of the protein binding signal. Average profiles of the RNA-seq signal were calculated as the mean level. The standard error appears on the graphs as a lighter area around the main line of the profiles. Abbreviations: TSS: transcription start site; TES: transcription end site.

**Figure 5 ijms-23-10662-f005:**
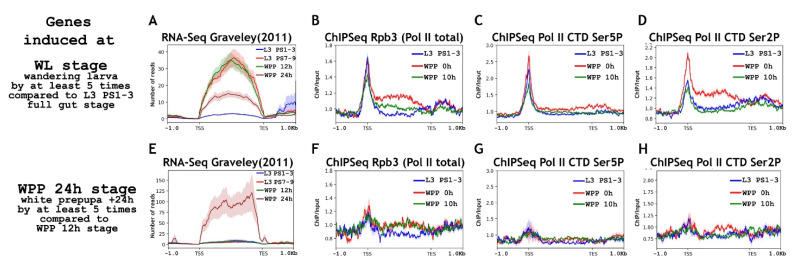
Average binding profiles of various Pol II isoforms on genes induced during *Drosophila* puparium formation (WL: wandering larva) and 24 h after puparium formation (WPP 24 h: white prepupa 24 h). Average of the (**A**) RNA-seq signal and an average distribution of (**B**) Rpb3 (Pol II subunit), (**C**) Pol II Ser5P CTD, and (**D**) Pol II Ser2P CTD binding across the genes induced in wandering larvae L3 (WL stage: wandering larva) (total 303 genes/852 transcripts). Average of the (**E**) RNA-seq signal and an average distribution of (**F**) Rpb3 (Pol II subunit), (**G**) Pol II Ser5P CTD, and (**H**) Pol II Ser2P CTD binding across the genes induced 24 h after puparium formation (WPP 24 h stage: white prepupa +24 h) (total 62 genes/108 transcripts). Pools of “WL genes” and “WPP 24 h genes” were selected according to the common rule, to be induced at least five times compared with “L3 PS1-3 full gut” and “WPP 12 h APF” stages, respectively. ChIP-Seqs were performed on the material of whole larvae of L3 PS1-3 full gut stage (blue line), prepupae 0 h after puparium formation (WPP, red line) and prepupae 10 h after puparium formation (WPP 10 h, green line). Protein binding levels were calculated as an enrichment (ratio of the corresponding ChIP-Seq signal to the input DNA). Average profiles were generated using the metagene mode (introns were ignored and gene bodies were scaled to 2 kb) and calculated as the median of the protein binding signal. Average profiles of the RNA-seq signal were calculated as the mean level. The standard error appears on the graphs as a lighter area around the main line of the profiles. Abbreviations: L3 PS1-3: full gut stage of larva L3; L3 PS7-9: empty gut stage of larva L3; WPP 0 h: white prepupa; WPP 10 h, 12 h, and 24 h: prepupa 10 h, 12 h and 24 h after puparium formation, respectively; TSS: transcription start site; TES: transcription end site.

**Figure 6 ijms-23-10662-f006:**
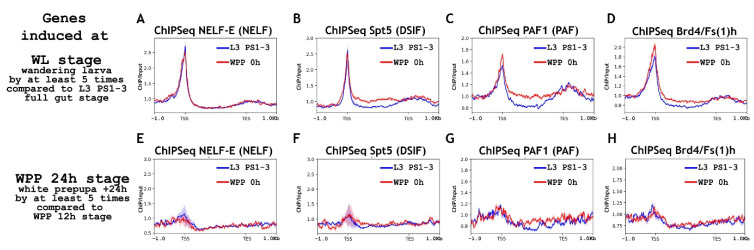
Average binding profiles of NELF-E (NELF), Spt5 (DSIF), PAF1 (PAF), and Bdr4/Fs(1)h on genes induced at *Drosophila* puparium formation (“WL” wandering larva) and 24 h after puparium formation (“WPP 24 h” white prepupa 24 h). Average distribution of (**A**) NELF-E (NELF complex), (**B**) Spt5 (DSIF complex), (**C**) PAF1 (PAF complex), and (**D**) Brd4/Fs(1)h binding across the genes induced in wandering larvae L3 (WL stage: wandering larva) (total 303 genes/852 transcripts). Average distribution of (**E**) NELF-E (NELF complex), (**F**) Spt5 (DSIF complex), (**G**) PAF1 (PAF complex), and (**H**) Brd4/Fs(1)h binding across the genes induced 24 h after puparium formation (WPP 24 h stage: white prepupa +24 h) (total 62 genes/108 transcripts). Pools of “WL genes” and “WPP 24 h genes” were selected according to the common rule, to be induced at least five times compared with “L3 PS1-3 full gut” and “WPP 12 h APF” stages, respectively. ChIP-Seqs were performed on the material of whole larvae of L3 PS1-3 full gut stage (blue line), white prepupae 0 h after puparium formation (WPP, red line) and prepupae 10 h after puparium formation (WPP 10 h, green line). Protein binding levels were calculated as an enrichment (ratio of the corresponding ChIP-Seq signal to the input DNA). Average profiles were generated using the metagene mode (introns were ignored and gene bodies were scaled to 2 kb) and calculated as the median of the protein binding signal. Average profiles of the RNA-seq signal were calculated as the mean level. The standard error appears on the graphs as a lighter area around the main line of the profiles. Abbreviations: L3 PS1-3: full gut stage of larva L3; WPP 0 h: white prepupa; TSS: transcription start site; TES: transcription end site.

**Figure 7 ijms-23-10662-f007:**
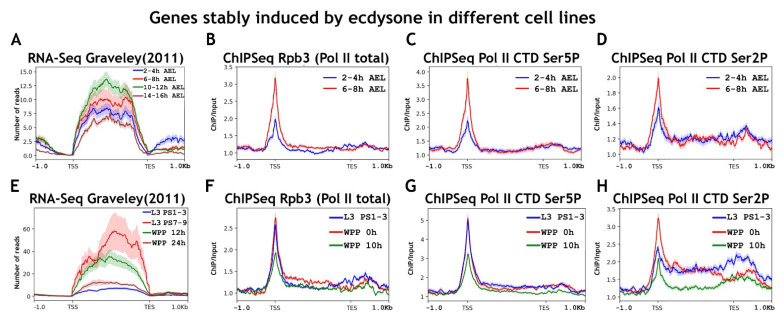
Average binding profiles of various Pol II isoforms on genes induced by ecdysone during *Drosophila* embryogenesis and metamorphosis average of the (**A**,**E**) RNA-seq signal and an average distribution of (**B**,**F**) Rpb3 (Pol II subunit); (**C**,**G**) Pol II Ser5P CTD; and (**D**,**H**) Pol II Ser2P CTD binding across the genes induced by 20-hydroxyecdysone (ecdysone) at various stages of *Drosophila* development (total 68 genes/236 transcripts). A pool of ecdysone-induced genes was selected using previously published data on the ecdysone response of various *Drosophila* cell lines [55]. ChIP-Seqs were performed on whole embryos of 2–4 h (blue line at (**A**–**D**)) and 6–8 h (red line at (**A**–**D**) after eggs laying (AEL); on whole larvae of L3 PS1-3 full gut stage (blue line on (**E**–**H**)), prepupae 0 h after puparium formation (WPP, red line on (**E**–**H**)) and prepupae 10 h after puparium formation (WPP 10 h, green line on (**E**–**H**)). Protein binding levels were calculated as an enrichment (ratio of the corresponding ChIP-Seq signal to the input DNA). Average profiles were generated using the metagene mode (introns were ignored and gene bodies were scaled to 2 kb) and calculated as the mean of the protein binding signal. The standard error appears on the graphs as a lighter area around the main line of the profiles. Abbreviations: L3 PS1-3: full gut stage of larva L3; L3 PS7-9: empty gut stage of larva L3; WPP 0 h: white prepupa; WPP 10 h, 12 h, and 24 h: prepupa 10 h, 12 h, and 24 h after puparium formation, respectively; TSS: transcription start site; TES: transcription end site.

**Figure 8 ijms-23-10662-f008:**
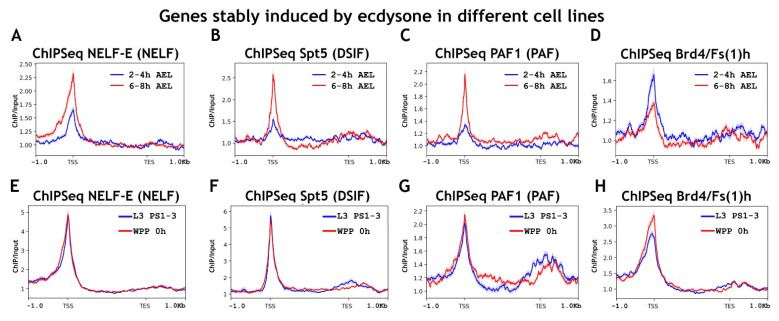
Average binding profiles of NELF-E (NELF), Spt5 (DSIF), PAF1 (PAF), and Brd4/Fs(1)h on genes induced by ecdysone during *Drosophila* embryogenesis and metamorphosis average distribution of (**A**,**E**) NELF-E (NELF complex); (**B**,**F**) Spt5 (DSIF complex); (**C**,**G**) PAF1 (PAF complex); and (**D**,**H**) Brd4/Fs(1)h binding across the genes induced by 20-hydroxyecdysone (ecdysone) at various stages of *Drosophila* development (total 68 genes/236 transcripts). A pool of ecdysone-induced genes was selected using previously published data on the ecdysone response of various *Drosophila* cell lines [55]. ChIP-Seqs were performed on whole embryos of 2–4 h (blue line at (**A**–**D**)) and 6–8 h (red line at (**A**–**D**) after eggs laying (AEL); on whole larvae of L3 PS1–3 full gut stage (blue line on (**E**–**H**)), prepupae 0 h after puparium formation (WPP, red line on (**E**–**H**)). Protein binding levels were calculated as an enrichment (ratio of the corresponding ChIP-Seq signal to the input DNA). Average profiles were generated using the metagene mode (introns were ignored and gene bodies were scaled to 2 kb) and calculated as the mean of the protein binding signal. The standard error appears on the graphs as a lighter area around the main line of the profiles. Abbreviations: L3 PS1-3: full gut stage of larva L3; WPP 0 h: white prepupa; TSS: transcription start site; TES: transcription end site.

**Figure 9 ijms-23-10662-f009:**
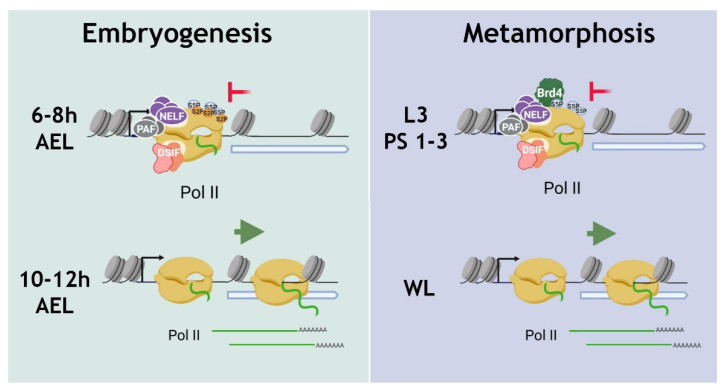
Schematic model describing specific features of RNA polymerase II “pause” complexes that prepare *Drosophila* genes induced at 10–12 h AEL of embryogenesis and during L3 PS7-9 empty gut stage of the metamorphosis (in wandering larvae: WL) for upcoming transcription during embryogenesis and metamorphosis. The “pause” of RNA polymerase II which prepares “10–12 h genes” for the induction during mid-embryogenesis is characterized by phosphorylation of its Pol II CTD not only by Ser5 but also by Ser2. The embryonic “pause” complex contains NELF, DSIF, and PAF complexes but lacks Brd4/Fs(1)h due to the low expression level of this protein at this stage of development. During metamorphosis, the genes, induced in wandering larvae (WL) use promoter-proximal pausing, which is characterized by a high level of Pol II CTD Ser5P and a low degree of Pol II CTD Ser2 phosphorylation. The “pause” complex contains NELF, DSIF, and PAF complexes and Brd4/Fs(1)h protein. The figure was drawn by the authors using *biorender.com* software (accessed on 10 August 2022).

## Data Availability

All obtained ChIP-Seq data were deposited into the Gene Expression Omnibus—GSE210971.

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
