# Peer review of "RNA Polymerase II “Pause” Prepares Promoters for Upcoming Transcription during Drosophila Development"

_ijms, 2022, doi:10.3390/ijms231810662_

Round 1

Reviewer 1 Report

The authors aim is to show that RNA polymerase II is poised in the TSS before transcription starts. For this the authors utilized ChIP-Seq analysis of various transcription factors in different developmental staged in Drosophila. The authors presented a great amount of NGS data that will be publicly available. In general, the authors should explicitly explain why they have chosen some factors over others and how they relate biologically to their results.

In the introduction, please add if Drosophila’s Rpb1 CTD contains heptapeptide repeats and if it is similar to other metazoan. Also explain how the addition of different PTMs in the CTD are specific to different functions related to transcription.

Please describe in text or new figure, how many subunits does the RNA pol II of Drosophila have?

Please explain the known function of each of these factors: NELF, DSIF, p-TEFb and is Brd4/Fs(1)h? For example is Brd4 a bromodomain containing protein? What is this recognizing if known? Is it recognizing DNA via an ET or a PTM via is BRD. How is this factor close to the TSS? What are the characteristics of these locations in the genome? How are these factors relevant to this work?

In introduction, mention examples of the developmental genes characterized by a synchronous pattern of transcription activation and some ecdysone-induced genes, probably the genes you chosen in Results.

Define various forms of RNA polymerase II

Explain how NELF-E, Spt5, PAF1 and Brd4/Fs(1)h regulate transcription

Explain why you have chosen the embryogenesis (2-4 hours and 6-8 hours AEL – after eggs laying) and three points of metamorphosis for you ChIP-Seq analysis. Are specific biological processes happening here?

Figure 1 does not show a result specifically. Please use a figure that reflects a result. Perhaps modify S1 and merge. Figure S1 is missing statistical analysis displaying that the compared data does not show significant changes.

“The selection of this set of genes was performed using previously published data by Graveley and colleagues on gene transcription at different stages of Drosophila development (Graveley et al., 2011)."

For all figures, please include a supplementary tables of these genes in your supplementary data. Include, how many genes does Drosophila have? How many genes did you choose for each analysis? Please clarify explicitly in all figures that this metagene plots are only from a set of genes that follow certain expression patterns. From the chosen list, show the amount of genes that have the plots pattern and how many genes do not have these pattern. Show gene models of selected genes that follow the patterns peaks close to the TSS with the different data sets. Explain figure captions in more detail.

"We observe the presence of some RNA polymerase II and elongation regulators at the 2-4h AEL stage, preceding the transcriptional activation stage of the 6-8h AEL genes pool." Where is this data? What are these genes? Needs supplementary table.

All figures, label the panels with A, B, C, etc. and describe in caption

In all figures, show some analysis that integrates the data of these plots to quantify what genes have the similar enrichment of the chosen factors use for ChIP. Discuss the results.

517, modify to New Generation to Next Generation

Methods, expand to what software you used to align, did you use BWA, HOMER, etc? Did you use deeptools? Please explain/expand.

Author Response

The authors aim is to show that RNA polymerase II is poised in the TSS before transcription starts. For this the authors utilized ChIP-Seq analysis of various transcription factors in different developmental staged in Drosophila. The authors presented a great amount of NGS data that will be publicly available. In general, the authors should explicitly explain why they have chosen some factors over others and how they relate biologically to their results.

We would like to thank the Reviewer for His/Her valuable suggestions, which really helped to improve our manuscript and gave it biological significance. We are particularly grateful for the idea of performing gene/transcript clustering to demonstrate which part of the regions fit the described model.

In the introduction, please add if Drosophila’s Rpb1 CTD contains heptapeptide repeats and if it is similar to other metazoan. Also explain how the addition of different PTMs in the CTD are specific to different functions related to transcription.

We have significantly expanded the part in the introduction describing the specifics of Drosophila RNA polymerase II. We also linked PTMs in the Pol II CTD with various stages of transcriptional cycle.

Please describe in text or new figure, how many subunits does the RNA pol II of Drosophila have?

We have included this information in the introduction. Drosophila Pol II RNA consists of 13 subunits, like Pol II of other Metazoans.

Please explain the known function of each of these factors: NELF, DSIF, p-TEFb and is Brd4/Fs(1)h? For example is Brd4 a bromodomain containing protein? What is this recognizing if known? Is it recognizing DNA via an ET or a PTM via is BRD. How is this factor close to the TSS? What are the characteristics of these locations in the genome? How are these factors relevant to this work?

We have significantly expanded the description of NELF, DSIF, p-TEFb and Brd4/Fs(1)h in the introduction section. In addition, at the suggestion of Reviewer II, we provided a schema describing the mechanisms of the various types of Pol II "pause" (presented in FigS1), which helped us further characterize these factors and their role in controlling Pol II elongation.

Indeed, Brd4/Fs(1)h is a bromodomains-containing protein capable of binding to acetylated histones and its most described role in transcription is the recruiting the active p-TEFb, releasing promoter-proximal pausing (Gaub et al., 2020; Jang et al., 2005). But the detailed mechanisms of Brd4/Fs(1)h recruitment are still in question, as it has been found that bromodomain inhibitors (such as JQ1) do not completely remove this protein from chromatin. This may be especially important for Drosophila Brd4/Fs(1)h, since the Drosophila Brd4/Fs(1)h has isoforms with different properties and different binding patterns in the genome (Cubeñas-Potts et al., 2017; Kuroda et al., 2020).

We incorporated the detailed characterization of Brd4/Fs(1)h recruitment and functions into the Introduction.

In introduction, mention examples of the developmental genes characterized by a synchronous pattern of transcription activation and some ecdysone-induced genes, probably the genes you chosen in Results.

We have mentioned both several developmental genes that are synchronously activated during Drosophila embryogenesis and some ecdysone-induced genes that are activated upon an increase in ecdysone titer.

Define various forms of RNA polymerase II

In the introduction, we have characterized several PTMs of the Pol II CTD and the enzymes that incorporate them.

Explain how NELF-E, Spt5, PAF1 and Brd4/Fs(1)h regulate transcription

These proteins are subunits of the factors controlling elongation NELF, DSIF, PAF, and Brd4/Fs(1)h, and we associate the functions of these proteins with the functions of the complexes they are incorporated. And we significantly expanded the description of the properties of the NELF, DSIF, PAF and Brd4/Fs(1)h complexes in the introduction of the revised manuscript.

Explain why you have chosen the embryogenesis (2-4 hours and 6-8 hours AEL – after eggs laying) and three points of metamorphosis for you ChIP-Seq analysis. Are specific biological processes happening here?

Investigations focusing on the development of organisms often encounter a problem, namely, difficulties in obtaining material that is highly synchronized in terms of the stage of development. A general idea in investigations of developmental stages is to collect an organism during the most synchronized moment, give it time to develop, and then collect the material. It must be noted that the more time that passes from the moment the material was collected, the more desynchronized the obtained population becomes.

We specifically decided to study early and middle embryogenesis in order not to move far from the point of synchronization, i.e., egg laying. In addition, important processes of embryogenesis occur at the following stages: at the 2-4h stage, active transcription of the genome begins for the first time, and at the 6-8h stage, specification and determination of cell fate occurs.  This is why these embryonic stages are often chosen by researchers (Ghavi-Helm et al., 2014; McKay and Lieb, 2013). We also chose these stages of embryogenesis so that other could correlate their data with ours.

Drosophila development has an additional stage, i.e., metamorphosis, which allows one to obtain highly synchronized material. Puparium formation has a clear set of markers at each stage, which makes it possible to obtain specific material (and the most synchronized stage for the collection is white prepupa stage). Drosophila prepupa formation is one of the key developmental stages when organism completely restructured – gross number of organs undergoes autophagy and apoptosis and the new ones are building of the specific larval structures called imaginal discs.

Figure 1 does not show a result specifically. Please use a figure that reflects a result. Perhaps modify S1 and merge. Figure S1 is missing statistical analysis displaying that the compared data does not show significant changes.

We have modified Figure 1 and merged it with Figure 1. Now it contains only statistically analyzed data. In order not to lose information and to correlate our data with known information, we marked the stages of development demonstrating the highest level of transcription of the corresponding genes in accordance with the flybase.org by asterisks.

“The selection of this set of genes was performed using previously published data by Graveley and colleagues on gene transcription at different stages of Drosophila development (Graveley et al., 2011)."

In the results, we described in more detail the principle by which the selection of genes was carried out.

For all figures, please include a supplementary tables of these genes in your supplementary data. Include, how many genes does Drosophila have? How many genes did you choose for each analysis? Please clarify explicitly in all figures that this metagene plots are only from a set of genes that follow certain expression patterns. From the chosen list, show the amount of genes that have the plots pattern and how many genes do not have these pattern. Show gene models of selected genes that follow the patterns peaks close to the TSS with the different data sets. Explain figure captions in more detail.

We have included tables, describing selected sets of genes, that can be used to generate .bed files (provided in the Supplementary). These tables contain the coordinates of all sets of transcripts used by us for analysis (three of them also contain information about which cluster this transcript belonged to in the cluster analysis).

We indicated in the text that for the selection of genes sets we used the information from Supplementary table 9 of the study (Graveley et al., 2011). The referred table contains 15139 genes (which we accept as a total number of Drosophila genes).

In the results, we described the number of genes and their corresponding transcripts that were selected by us for each analyzed set.

In all the figures, we tried to clarify in more detail the principles for selecting the analyzed sets of genes/transcripts.

To reveal which part of the analyzed genes/transcripts demonstrate the pattern with the “paused” Pol II in sets of genes induced at the 6-8h and 10-12h of embryogenesis and L3 PS7-9 Empty gut stage of metamorphosis we have performed a clustering analysis based on the amount of total Pol II bound to the promoters (these results are provided in FigS2, FigS3 and FigS6). This made it possible to demonstrate that only a part of the genes/transcripts show binding patterns corresponding to the averaged profiles. We explain this by the limited sensitivity of the ChIP-Seq method, as well as by the fact that the ChIP-Seqs performed on the material of a whole embryo or larva cannot demonstrate the state of gene promoters working in a small fraction of the cells of the analyzed organism.

The bright side of the results of our cluster analysis is the result demonstrating a correlation in the behavior of all the transcription factors we studied during clustering. Those sites where we detected the presence of Pol II also demonstrated the binding of all other transcription factors in accordance with the model (with the exception of Brd4/Fs(1)h, which simply did not demonstrate the binding to all promoters at the stage of 6–8 h of embryogenesis).

Gene models of selected genes that follow the “paused” patterns (corresponding to cluster 1 in all sets where cauterization was performed) are provided in FigS2, FigS3 and FigS6 and described in the Results.

"We observe the presence of some RNA polymerase II and elongation regulators at the 2-4h AEL stage, preceding the transcriptional activation stage of the 6-8h AEL genes pool." Where is this data? What are these genes? Needs supplementary table.

These results are provided in Fig 2 and Fig S2 and described in the Results. Tables describing sets of the analyzed region were provided in the Supplementary.

All figures, label the panels with A, B, C, etc. and describe in caption

We have labelled all the panels in all the figures

 In all figures, show some analysis that integrates the data of these plots to quantify what genes have the similar enrichment of the chosen factors use for ChIP. Discuss the results.

We have performed a clustering analysis which let us to demonstrated that the presence of NELF, DSIF, PAF and Brd4/Fs(1) correlates with the presence of “paused” Pol II (these results are provided in FigS2, FigS3 and FigS6). Results of the analysis are described in the Result and Discussion sections.

517, modify to New Generation to Next Generation

We have replaced “New Generation” with “Next Generation” 

Methods, expand to what software you used to align, did you use BWA, HOMER, etc? Did you use deeptools? Please explain/expand.

We have actually described the software we used to align and prepare our raw data for the analysis (in the methods section), but we have expanded this section in revised manuscript as suggested by the reviewer.

Reviewer 2 Report

The work done by Mazina et al investigated the mechanism of the transcription activation from the aspect of PolII machinery recruitment and release at two developmental stages (embryogenesis and metamorphosis) using drosophila as cell model. The results are very interesting, data is sufficient and well discussed across the manuscripts. Below are additional comments that can help enhance the work further:

1. From line 36 to 96, the author described the multiple steps of transcription cycle (e.g. initiation and elongation) and the related proteins involved in different steps. Based on the polII binding and transcription activity, the authors define the post-initiation transcription regulation in three ways: PrPP, poised and “post-paused”. I appreciate that the authors tried to give a very clear illustration of the background of this study, however, that’s a long introduction. Could the author provide a figure (in the SI or merged with Fig. 9) to give the reader a clear visualization of the points in the long text?

2. What is the difference regarding the distance between the binding site of PolII and TSS at the three different states: PrPP, poised and “post-paused”?

3. “This observation suggests that temporal stimuli induce Pol II recruitment in both target and non-target tissues, while spatial stimuli affect the transcriptional elongation.” What does the “spatial stimuli” refer to?

4. “The pattern of changes in gene transcription in the analyzed tissues corresponded to 123 the expected (previously published).” Provide evidence (either data or citations);

5. Explain why the Spt5 is depleted from gene body when the transcription is activated (e.g. red curve in Figure 2);

6.The finding in terms of the different mechanisms of pausing used in embryogenesis and metamorphosis is interesting (line 380 to 389). Could the authors explain why the composition and properties of the paused PolII complexes differ in mid-embryogenesis and metamorphosis?

Author Response

The work done by Mazina et al investigated the mechanism of the transcription activation from the aspect of PolII machinery recruitment and release at two developmental stages (embryogenesis and metamorphosis) using drosophila as cell model. The results are very interesting, data is sufficient and well discussed across the manuscripts. Below are additional comments that can help enhance the work further:

We are very grateful to the Reviewer for the evaluation of our manuscript and for the valuable suggestions, which, in our opinion, really helped to improve the manuscript.

  1. From line 36 to 96, the author described the multiple steps of transcription cycle (e.g. initiation and elongation) and the related proteins involved in different steps. Based on the polII binding and transcription activity, the authors define the post-initiation transcription regulation in three ways: PrPP, poised and “post-paused”. I appreciate that the authors tried to give a very clear illustration of the background of this study, however, that’s a long introduction. Could the author provide a figure (in the SI or merged with Fig. 9) to give the reader a clear visualization of the points in the long text?

We thank the reviewer for the suggestion to add a scheme describing various ways to control the elongation of the RNA polymerase II - we have placed such a scheme in the manuscript as FigS1.

We had to further expand our Introduction and describe transcriptional cycle, Pol II isoforms and elongation control factors in more detail as was suggested by Reviewer 1. We hope that the scheme in the FigS1 helps to get the main ideas of the Introduction.

  1. What is the difference regarding the distance between the binding site of PolII and TSS at the three different states: PrPP, poised and “post-paused”?

We believe that right now there is not enough data to answer this question with absolute certainty. The best-described Pol II "pause" state is PrPP, and it has been demonstrated that in this state, Pol II stalls after synthesis of 20-50 nucleotides of RNA, in other words, several tens of nucleotides downstream the promoter (or a little closer to the TSS - in the case where was a backtracking) (https://doi.org/10.1038/s41580-020-00308-8). 

The states of “poised” and “post-paused” Pol II are much less characterized, so there is less certainty in the position of the Pol II at the gene. Since the “poised” state occurs before any RNA synthesis is initiated, it is assumed that the Pol II in this state is associated exactly with the promoter and does not leave it. It is assumed that one of the reasons for the stalling of the Pol II CTD Ser2Phospho at the promoter is the state of chromatin, therefore the position of the Pol II in the “post-pause” state depends on the position of the +1nucleosome and may be different for various genes (10.1016/j.molcel.2020.02.014).

We have tried to include information describing the "pause" position of Pol II in the Fig. S1.

  1. “This observation suggests that temporal stimuli induce Pol II recruitment in both target and non-target tissues, while spatial stimuli affect the transcriptional elongation.” What does the “spatial stimuli” refer to?

By "spatial" transcription regulation signals we mean master-regulators and morphogens. We believe that tissue-specific transcription factors can also be classified as “spatial stimuli”.

We have added the necessary clarification to the text of the article.

  1. “The pattern of changes in gene transcription in the analyzed tissues corresponded to 123 the expected (previously published).” Provide evidence (either data or citations);

We have placed the required link to the text of the article.

  1. Explain why the Spt5 is depleted from gene body when the transcription is activated (e.g. red curve in Figure 2);

We would like to note that we were also surprised to see such a distribution of Spt5 across the genes bodies in embryogenesis. We believe that at least two important aspects contribute to this fact.

  • We guess that when moving to the gene bodies, the structure of the Pol II complex may change and, in the course of these changes, Spt5 may move away from DNA, which reduces the likelihood of its cross-linking by the formaldehyde. This explanation is supported by the fact that even in metamorphosis, where we still see the presence of Spt5 in the body of the gene, its level is still lower than on the promoter.

  • But still in embryogenesis we find a lower level of Spt5 on gene bodies than in metamorphosis. We associate such a special behavior of Spt5 in embryogenesis with a specific set of proteins associated with the Pol II at this stage - in particular, the absence of Brd4/Fs(1)h, but quite possibly something else. Perhaps due to the absence of Brd4/Fs(1)h, p-TEFb is inefficiently recruited to genes, and as a result there is no effective phosphorylation of Spt5. Unphosphorylated Stp5 poorly associates with the Pol II and as a result it does not pass into the elongation phase together with the Pol II.

Our observation of the presence of Pol II CTD Ser2P on paused promoters during embryogenesis may seem contradictory to our explanation (this is strange considering that these genes may have problems with p-TEFb recruitment due to the absence of Brd4).

We associate the presence of Pol II CTDSer2P on the promoters in the embryogenesis not with increased activity of p-TEFb, but with problems in the formation of a productive elongation complex by Pol II - the increase in Pol II CTDSer2P modification occurs not because the p-TEFb complex is actively working, but because the Pol II cannot enter the gene body. This argumentation is supported by data from our recent article on the role of NELF in the transcription of ecdysone genes (10.1038/s41598-020-80650-1) - there we showed that RNAi of NELF subunits leads to a disruption in the interaction between Spt5 and Pol II and a very strange consequence - a decrease in the total level of Pol II on the promoters, but at the same time an increase the degree of its phosphorylation by Ser2.

All of these data very well indicate that there are some other forces that keep the phosphorylated Pol II Ser2 in the proximal part of the promoter. The truth is, it's still not clear what it is.

6.The finding in terms of the different mechanisms of pausing used in embryogenesis and metamorphosis is interesting (line 380 to 389). Could the authors explain why the composition and properties of the paused PolII complexes differ in mid-embryogenesis and metamorphosis?

So far, the key explanation we can draw from the results described in this article is the absence of Brd4/Fs(1)h in mid-embryogenesis. We believe that in the absence of Brd4/Fs(1)h, p-TEFb recruitment processes may be disrupted and this may result in an inefficient assembly of the Pol II elongation complex. Examining the expression profiles of Brd4/Fs(1)h during embryogenesis on the Western blots (FigS5), we saw that there is a certain gap in the expression of this protein in early embryogenesis. We attribute this to the delayed onset of zygotic expression of Brd4/Fs(1)h. It is likely that the gene encoding Brd4/Fs(1)h is induced not in the main wave of ZGA in embryogenesis, but later.

We assumed that not only Brd4/Fs(1)h may absent at this stage of the development (in mid-embryogenesis).

Now in our laboratory we are working on this phenomenon further - in particular, we have obtained antibodies to cycT and looked at how this protein is expressed in embryogenesis (unfortunately, we have not succeeded in obtaining the antibodies to cdk9).

But even with cycT, we got an interesting result - we see that at those stages of embryogenesis where Brd4/Fs(1)h is absent, cycT is also absent. That is, it seems that Drosophila lacks p-TEFb in mid-embryogenesis.

By the way, this result correlates well with what we see when stain western blots with antibodies against total Pol II – RPII215 (Fig S5). The lower unphosphorylated form of RPII215 is stably present at all stages of embryogenesis, while the upper phosphorylated form disappears for a while, as well as Brd4/Fs(1)h and CycT. So really p-TEFb at this stage of development is not active enough.

Unfortunately, at present, we are not yet ready to publish the results describing the expression of other elongation regulators in embryogenesis. These are preliminary results so far. We still believe that the main message of our article is sufficiently supported by experimental data.

Round 2

Reviewer 1 Report

Fig S4 is presented before FigS3, please fix

Tables must be indexed in text

Author Response

Fig S4 is presented before FigS3, please fix 

We double-checked the correct order of the figures in the text. We found it correct but we noticed a typo mistake in the reference to Fig S3 which we fixed. Now all the figures have the correct order of appearance in the text.

Tables must be indexed in text

All Supplementary tables were indexed in the text in the order of their appearance.

We would like to thank the Reviewer for His/Her helpful comments on our manuscript.